# Virtual Versus Physical Number Line Training for 6-Year-Olds: Similar Learning Outcomes, Different Pathways

Eva-Maria Ternblad *, Maybi Morell Ruiz and Sonja Holmer

Department of Cognitive Science, Lund University, SE-22100 Lund, Sweden;
maybi.morell_ruiz@lucs.lu.se (M.M.R.); sonja.holmer@lucs.lu.se (S.H.)
* Correspondence: eva-maria.ternblad@lucs.lu.se

**Abstract**

According to previous research, young children's numeracy skills may be scaffolded by practicing on the number line. A number line estimation task (NLET) is often conducted with pen and paper, while linear number games are often implemented on a computer or a tablet. If and how the format—physical or digital—influences the accuracy of the estimations is, however, not well-known. If regarding NLET performance as dependent on specific strategies and hypothesizing that these strategies may be affected by the material used, we may also assume that different materials may either support or hinder children's learning. In this paper, we explore whether training with a physical versus a virtual NLET game will affect children's strategies when solving NLETs, and if these strategies relate to the accuracy of the estimations. Sixty-two 6-year-old children played an NLET game (virtual or physical) for three sessions, being scaffolded and guided by a researcher. NLET performance was measured by pre- and post-tests, as well as during the intervention. The results show that even if the condition did not significantly affect the children's overall numeracy skills, the children in the physical condition did express more advanced strategies during the intervention. These strategies, in turn, predicted NLET performance.

**Keywords:** early numeracy; number line estimation; NLET strategies; physical and virtual interaction

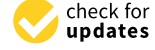

## 1. Introduction

Early numerical skills have been shown to be one of the strongest predictors for future academic achievement—not only in math, but also with respect to succeeding in school in general (Duncan et al., 2007; Geary, 2011). Consequently, ensuring that young children acquire basic knowledge about symbolic and non-symbolic representations of numerical magnitudes is crucial, both for improving the life quality of individuals and for society as a whole (see also Parsons & Bynner, 2005).

A well-known pedagogical activity for strengthening children's numeracy skills is to play linear number games (Siegler & Ramani, 2009). Another—although a slightly different exercise—is to estimate numbers on a number line—often referred to as number line task estimation (NLET) (Siegler & Opfer, 2003). Evidently, by visualizing and ordering numbers on a straight line, and by letting the distances between the numbers reflect the numbers' reciprocal relations, it is possible to scaffold both younger and older children's understanding of numerical magnitude and ordinality.

Linear number games as well as NLETs exist in both digital and physical formats. Whether one can expect the same learning outcomes from both, however, is not well-known.

As of today, there is a growing body of research comparing physical versus virtual learning materials, indicating that while virtual interaction can be beneficial in some cases, physical interaction may be more favourable in others (see for instance Brinson, 2015; M. A. Rau, 2020). Research evaluating embodied aspects of numerical cognition and learning also show a strong relationship between children's interactions and their actual performance, stating that limiting their interaction may decrease their ability to conduct numerical operations (Gordon & Ramani, 2021; Manches et al., 2010). When it comes to linear number games and NLETs, however, such comparisons are rare.

Moreover, even though some meta-analyses of conducted NLET studies indicate differences in performance measures depending on the medium used (Ellis et al., 2021), other studies show that assessments of number estimations by digital or pen- and paper-based tools yield similar results (Piatt et al., 2016). The study presented in this paper addresses this research gap. This is conducted by comparing 6-year olds' learning with a digital NLET game to learning with a pen- and paper-based NLET game. The goal with the study is not only to evaluate potential differences in learning outcomes, but also to see if and how the learning environment may affect beneficial and valuable strategies—such as interacting with the material and using multiple reference points along the line.

### 1.1. Number Line Estimations and Number Line Games

#### 1.1.1. The Number Line Estimation Task (NLET)

The number line estimation task (NLET) was first introduced by the cognitive psychologist Robert Siegler (Siegler & Opfer, 2003), who developed the task to study how children and adults understand numerical magnitudes. In an NLET, the task is to estimate a target number on a (partially) empty number line with a specific range (such as 0–10, 0–20, or 0–100). Performance on NLETs has been shown to be strongly positively correlated with a series of other mathematical competencies (see for instance Schneider et al., 2018; Siegler, 2016).

The most commonly used NLET format is the *bounded* number line, where the number line is presented as a straight horizontal line with reference points in the beginning and the end (Siegler & Opfer, 2003) (other formats also exists, see for instance Cohen & Blanc-Goldhammer, 2011). In the bounded NLET condition, the two reference points often cover a numerical interval of even tens (as the ones described above) but other intervals are of course possible to use. There is today a large body of research making use of NLETs, both for assessing children's and adults' mathematical competencies (Ellis et al., 2021; Schneider et al., 2018) and for studying processes underlying numerical cognition (H. C. Barth & Paladino, 2011; Izard & Dehaene, 2008; Siegler, 2016).

#### 1.1.2. Theoretical Perspectives on Number Estimation Abilities

The NLET does not only provide information about children's or adults' mathematical abilities—it also says something about the capacity to estimate numbers and magnitudes. Over the past decades, this, along with a growing interest in children's cognitive and numerical development, has resulted in a series of empirically supported theories on how human numerical abilities evolve—from making approximate estimations of smaller numbers to engaging in more complex numerical processing. The most well-known of these is perhaps the concept of a mental number line, where numerical magnitudes are arranged from left to right, and where numerical representations initially are organized logarithmically, but later shift to being spaced linearly (Berteletti et al., 2010; Siegler & Opfer, 2003).

A contrasting view of how to interpret the results from NLET studies, however, is to take the spatial properties of the number line into account, as well as to regard the

estimations as a result of more or less explicit strategies. As an example, Slusser and colleagues (2013) propose that the accuracy on the NLET can be explained by proportional reasoning, and that it is the gradual understanding of numbers and proportions (spatial as well as numerical) that finally refines the estimations (see also H. C. Barth & Paladino, 2011). A similar idea has been put forward by Petitto (1990), who studied how younger and older children (in grade 1 to 3) utilized the range of a bounded number line (0–10) in different ways. While the first graders were more inclined to start from one of the endpoints of the number line—counting from 0 and forward, or from 10 and downward—the third graders often used other reference points to calibrate their estimations (such as the midpoint, or both beginning and endpoints). These results have later been replicated in a series of studies (H. Barth et al., 2016; Peeters et al., 2016; Schneider et al., 2008; Simon & Schindler, 2022; van't Noordende et al., 2016).

White and Szűcs (2012), however, adopt a more balanced position: while their findings support a developmental shift from logarithmic to linear patterns in NLET accuracy between first and second graders, they also admit that children's evolving strategies contribute to changes in their NLET performance. Without committing to a specific theory about how the number line concept is mentalized and represented in the human mind, the present study does assume—based on the research presented in the passages above—that performance on the NLET is strongly related to the employed strategies, such as using proportional reasoning, or using different reference points along the line. We therefore also expect that general numerical abilities—such as counting forwards and backwards and understanding relationships between numerals—will influence the results. This, in turn, means that we believe the estimation component in the NLET to be less prominent.

### 1.1.3. NLET as a Training Tool

While using NLET as an assessment tool is common, utilizing it as a training tool—at least in its original form—is slightly rarer. Some research actually questions the benefit of NLET training, revealing that improvements on the NLET transfers poorly to other domains, such as number comparison or arithmetic (Lunardon et al., 2023; Maertens et al., 2016; Wei et al., 2023). Still, there are studies indicating that practicing NLET in lower grades may significantly improve children's numerical competencies (H. Barth et al., 2016; Kucian et al., 2011; Opfer & Siegler, 2007; Sella et al., 2021; A. J. Wilson et al., 2009). Practicing on the NLET has also been shown to strengthen children's remembrance of numerals (Thompson & Opfer, 2016).

In this study, we put forward the idea that the number line can serve as a useful instrument for learning to evaluate and compare numbers and magnitudes, which in turn may strengthen other mathematical abilities. Moreover, we also point out that the specific type of training may heavily influence what is learned. As shown in a study by H. Barth et al. (2016), local corrective feedback can have a significant impact on both estimation patterns and NLET accuracy. We therefore consider the guidance and support provided to the children during NLET training—such as directing their attention to different points on the line and providing immediate informative feedback—to be important. In addition, we also emphasize that NLET training will benefit from a learning material that scaffolds and facilitates the children's interactions with the number line as such, helping them to apply useful strategies and helping them to understand how different numbers relate to one another.

*1.2. Physical Versus Virtual Learning Environments*

1.2.1. The Embodied and Distributed Nature of Solving Numerical Tasks

Even if mathematical operations may be seen as mental activities, they are seldom conducted in a vacuum. On the contrary, humans tend to utilize all conceivable and available tools when solving mathematical problems (see for instance M. Wilson & Clark, 2008). The embodied nature of mathematical thinking is explicitly obvious in children, who tend to rely on a series of strategies to both acquire and express numerical knowledge. The most prominent of these is perhaps finger counting, a technique that is spontaneously used by young children all over the world (Bender & Beller, 2012). Other common strategies (also used by adults) are pointing at objects when counting them (Alibali & DiRusso, 1999), utilizing space to arrange objects to be counted (Gärdenfors & Quinon, 2021), and gesturing when reasoning about spatial relations (Nathan et al., 2021). There is also a substantial body of research showing that restricting children's interactions when calculating or reasoning about numbers may impair their performance (Goldin-Meadow et al., 2001; Manches et al., 2010).

From a cognitive perspective, making use of embodied strategies or routines—as well as of external tools—is often explained as a way of off-loading working memory resources (Barsalou, 1999; Kirsh, 1995). Given that working memory in early childhood is limited—even though it undergoes a substantial development around age six through adolescence (Cowan & Alloway, 2008; Gathercole et al., 2004)—it is not surprising that such strategies are particularly important for younger children. Recent studies also reveal children's counting abilities to be significantly related to working memory development, with counting backwards having a substantially slower and later development trajectory than counting forwards (Ahmed et al., 2022; Reynolds et al., 2022).

The tendency of humans to make use of the external world as a means of thinking is sometimes referred to as distributed or embedded cognition, stating that it is impossible to separate internal processes from interactions with the outside world (Clark & Chalmers, 1998; M. Wilson, 2002). When it comes to teaching and learning, this means widening the unit of analysis from a single learner to the social and cultural aspects of learning, including peers, tutors, tools, and learning materials (Greeno et al., 1996; Sawyer & Greeno, 2009). Learning can thereby also be understood as the expansion of associations between memorized facts and procedures and perceived problems and tools, integrating both external and internal representations. Being slightly more moderate in their interpretation of such a paradigm, Schwartz and Goldstone (2015) choose to describe learning as being primarily about coordination—coordination between perception and action, coordination between symbolics and semantics, and coordination between memorization and understanding. Since our primary interest is to explore the learning materials' impact on the children's interactive patterns—as well as their learning outcomes—we believe this perspective to be appropriate for the present study.

1.2.2. Learning Math with Physical or Virtual Representations and Tools

Given that the digitalization of educational settings is one of the most ground-breaking societal changes during recent decades, we have chosen to compare a virtual learning material with a pen- and paper-based one. Digital technology has not only replaced the use of pen and paper with interactions with mouse, keyboard, and touchscreen, but it has also made it possible to replace static materials, such as textbooks, with multimodal interactive interfaces. These tools can provide students with individualized support and immediate feedback, while also embedding routine tasks—such as arithmetic problems—within game-like narratives aimed at increasing student motivation and engagement.

The digitalization of learning materials has been particularly pronounced within STEM subjects; however, whether this development has led to improved learning outcomes is still unclear. A series of meta-analyses reveal that while computer-based STEM learning environments may be beneficial in some cases, traditional physical tools may be more useful in others (Brinson, 2015; Gui et al., 2023; M. A. Rau, 2020). When it comes to math, comparative studies also show heterogeneous findings. Even if multiple meta-analyses indicate that digital game-based learning in mathematics can be highly effective—particularly when compared to traditional teaching methods (Hussein et al., 2022; Fadda et al., 2022)—studies specifically examining two versions of a learning material reveal more nuanced results. To give an example, a study on primary school students' use of virtual-based ten blocks showed that these were just as (or even more) effective for learning than physical cubes and rods (Litster et al., 2019). However, in another study, Nikiforidou (2019) compared tangible versus virtual manipulatives for preschoolers learning about probabilistic reasoning. The results here imply that the children showed a significantly better understanding of the likelihood of events when interacting with tangible rather than computer-based representations.

When it comes to the use of pen and paper versus computer-based interaction in math education, results also differ. Even if virtual interfaces can be supportive—especially in subjects like geometry (see for instance Juandi et al., 2021)—writing and drawing by hand can sometimes be preferable. In a study by Kop et al. (2020), it was shown that eighth graders who graphed formulas by hand when solving linear algebra problems outperformed students that did not. And when it comes to young children's learning and cognitive development, drawing by hand should not be underestimated. For example, Sinclair et al. (2018) conducted a study on the topic and found an intriguing, dynamic interplay between 6-year-olds' drawing of geometric figures and their visual perception, gestures, and language development.

### 1.2.3. Physical Versus Virtual NLETs

Even if NLET as an assessment tool is often conducted with pen and paper (e.g., H. C. Barth & Paladino, 2011; Ellis et al., 2021; Petitto, 1990; Slusser et al., 2013), number line training is often conducted through digital tools and games, where the number line, or other linear representations of numerals, is only one element of many (e.g., Sella et al., 2021; Siegler & Ramani, 2009; Whyte & Bull, 2008; A. J. Wilson et al., 2009). Still, direct comparisons between digital and physical NLETs appear to be scarce. To the authors' knowledge, only two studies have compared NLETs conducted with pen and paper vs. on a tablet or computer. The first one is a study by Piatt et al. (2016), who investigated whether sixth graders performed differently if they estimated target numbers on a virtual number line, using a stylus to make a mark, compared to if they took a traditional test with pen and paper. A total of 32 students estimated 20 target numbers on a bounded number line between 0 and 100 in two conditions (paper and tablet). The researchers evaluated the students' performance, but also their preferences and attitudes towards the two different conditions. No significant differences were found, but when given the option to choose what material to use, two thirds of the students chose the tablet.

The second study is a qualitative inquiry by Weng and Bouck (2016), who examined three students with intellectual disability performing number line training for 10 training sessions—5 with a paper-based and 5 with an app-based number line. Even if both materials were effective for the specific students, the app-based number line was slightly more effective—both in terms of accuracy and of completion time of number comparison tasks on a post-test.

1.2.4. Evaluating and Comparing Virtual and Physical Learning Environments

Although an increasing number of studies compare virtual and physical learning environments, we still lack a clear understanding of how the choice of materials shapes the learning situation—not least when it comes to younger children learning math. Most research in this area also relies primarily on outcome measures, such as post-test performance. However, prior work has shown that important mechanisms often unfold during the learning process, in students' gestures, discourse, and interaction patterns (e.g., M. Rau, 2017; M. A. Rau & Herder, 2021). Yet as Rau's review of 54 studies comparing virtual and physical materials in STEM subjects highlights (2020), such approaches remain comparatively rare, with the majority of studies focusing narrowly on test-based outcomes. Consequently, there is a need for studies using mixed methods to capture *how* learning happens in virtual versus physical learning settings, not just *that* it happens.

*1.3. Research Aims and Hypotheses*

Based on the theoretical and empirical background outlined above, we sought to examine whether working in a physical as opposed to a virtual learning environment would differentially affect children's NLET performance, depending on the properties and interactions provided by the learning environment itself. In the study at hand, 6-year-old children practiced estimating target numbers on a bounded empty number line ranging from 0 to 20. This was conducted by playing an engaging number line game, either on a tablet or with pen and paper, being guided and supervised by a researcher. The goal of the study was to evaluate whether these two versions of the learning material shaped the children's observable interaction patterns differently, and whether such patterns, in turn, influenced the accuracy of their estimations. More specifically, the following hypotheses were formulated:

**H1.**  *NLET training will improve the children's NLET abilities compared to a control group, and this improvement will be more pronounced for target numbers far from 0.*

**H2.**  *The type of learning environment (virtual/physical) will affect the children's NLET strategies during the intervention.*

**H3.**  *These strategies will, together with the type of learning environment (virtual/physical), affect the accuracy of the estimations when playing the game.*

**H4.**  *Both the type of learning environment (virtual/physical) and the learned strategies will influence the children's NLET performance on a post-test.*

Although the hypotheses specify expected links between learning environments, strategies, and performance, it remains unclear which learning material will be more effective. In addition, to ensure high ecological validity of learning environments, they are likely to differ in more than one way. This means that different learning materials may shape the learning situation in different ways, whereby aspects such as time spent per task or child/researcher interactions will not be entirely equivalent (this is a well-known problem when making comparisons between virtual and physical learning environments, see for instance M. Rau, 2017; M. A. Rau, 2020). Consequently, the study takes an explorative approach, focusing on how learning environments and strategies interact with training to shape children's NLET performance. This makes it possible to identify the strengths and limitations of each material without assuming one is superior. However, determining exactly what specific feature in each material drives potential effects will be more difficult to entangle.

## 2. Method

The work presented in this paper is a part of a larger study on children's numerical cognition, conducted in Sweden during the fall of 2023 with 145 kindergarten children. While the comprehensive study examines how NLET training affects both symbolic and non-symbolic numerical abilities—and the interplay between them—this study uses a subset of the dataset to specifically compare the effects of different types of learning materials on interaction patterns and NLET performance. Even if the participants in the two studies partly overlap, the analyses and the data points used in the analyses differ.

The study comprises a three-week intervention accompanied by pre- and post-tests. During this period, the students played an educational NLET game while being supervised and scaffolded by a researcher. All testing, and all NLET training, was conducted in a one-to-one setting. The students occasionally also received hints about proper strategies and were asked questions about them.

### 2.1. Materials

2.1.1. The Number Line Game

The NLET game was framed within a narrative in which animals were looking for food along a bounded empty number line ranging from 0 to 20. It consisted of three training rounds, each featuring 10 predefined target numbers, mixing smaller and larger values, with some consecutive target numbers presented one after another (see Appendix A). In each of the rounds, a character (a frog, a kangaroo, and a rabbit) was presented, asking the child for assistance. After the child's estimation, immediate feedback was provided, consisting of the child's estimation and the correct position of the target number together with a dish (pancakes, ice cream, spaghetti, etc.). The reason for not randomizing the target numbers was twofold: first, it would have been very difficult to randomize the target numbers in the physical condition, and second, we wanted to assure that a child who did not complete the game would have encountered both higher and lower numbers. Using pre-defined target numbers also facilitated the observation protocol (see further details in Section 2.2.1).

The game was created in two versions, one physical and one virtual, representing two experimental conditions. In the virtual condition, the children played the game on a tablet, and the narrative and characters were presented on screen—visually as well as orally (see Figure 1). As an example, the frog said, "Hi, I'm the frog George. I'm looking for food. The food should be somewhere along this line. Could you help me find it?". Next, the target number was presented—visually and orally—and the child estimated its position by clicking on the number line, or by swiping back and forth along it. Subsequently, after a fixed number of seconds, the game repeated the target number—visually and orally—and provided visual feedback. The character also commented on the response's accuracy by oral positive/negative feedback. That is, if the child's estimation was sufficiently accurate ($\leq 1$ unit), the virtual character provided positive feedback, saying something like, "Oh, yum, pancakes!". If the estimation had a lower accuracy, the character responded, "Oh, the food was over there...".

In the physical condition, the children solved the tasks with pen and paper. In this case, the narrative was presented by a researcher, and the characters were represented by plastic pen holders (see Figure 2). The researcher read the target numbers aloud, turning pages in a booklet with one target number per page. After the child's estimation, feedback was provided by placing a transparent sheet with the target number's correct position over the child's response. The researcher also commented orally on the estimation's accuracy. Since this condition was substantially slower due to practical issues with handling the physical materials, 15 target numbers were selected from the virtual condition, balancing

higher and lower numbers. A maximum time limit of approximately 12 min per session was also set for assuring substantial game play for all children using this material.

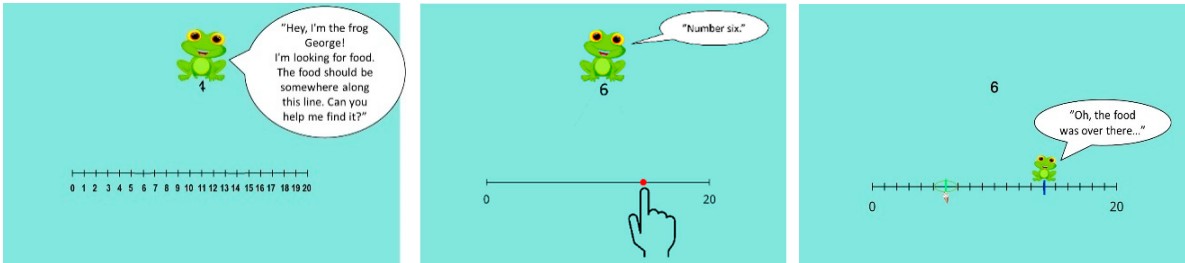

**Figure 1.** A character in the virtual NLET game gives an oral presentation of the narrative and the number line (**left**), introduces a target number (**middle**), and provides feedback (**right**).

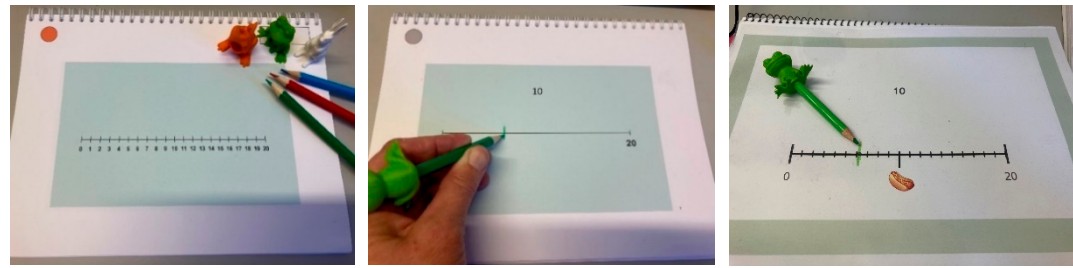

**Figure 2.** The physical NLET material and the number line (**left**), presentation of the target number together with the child's estimation (**middle**), and the feedback (**right**).

It should be noted that the two versions of the game were not designed to optimize the study's internal validity. Instead, we focused on ensuring that the experiment would have high ecological validity, and that the different learning materials would resemble commonly used educational and numerical games as closely as possible. The differences between conditions are presented in Table 1.

**Table 1.** Description of the NLET materials in the two conditions.

|  | Physical Condition | Virtual Condition |
|---|---|---|
| Materials | Booklet with printed number lines, coloured pencils, and plastic pen holders | Digital game<br>Tablet |
| Narrative | Delivered by a researcher (orally) | Delivered by the game (orally)<br>Commented by a researcher |
| Tasks | ~5 × 3 target numbers per session<br>Delivered, scaffolded and commented on by a researcher (orally + visually) | 10 × 3 target numbers per session<br>Delivered by the game (orally + visually)<br>Scaffolded/commented by a researcher |
| Student response | Making a mark with a pen<br>No time limit | Clicking/swiping on screen<br>No time limit |
| Feedback | Delivered by researcher (orally + visually)<br>No time limit | Delivered by the game (orally + visually)<br>Commented/elaborated on by a researcher<br>Fixed time limit (5–6 s) |
| Playing time | Limit set to 12 min | No time limit—ended as soon as the child completed the last task |

### 2.1.2. Pre-Tests and Post-Tests

The week before and the week after the intervention, pre- and post-tests were administered in a one-to-one setting. The test battery consisted of a series of tests in early numeracy and math: the Ani Banani Math Test (ABMT), the Numeracy Screener, and a digital NLET.

The Ani Banani Math Test (ABMT) is a Norwegian screening test that adopts a game-based format and evaluates arithmetic, geometry, and problem-solving skills (ten Braak & Størksen, 2021; Størksen & Mosvold, 2013), while the Numeracy Screener consists of a pen and paper assessment of symbolic and non-symbolic discrimination skills (Nosworthy et al., 2013). In this study, the total scores on the pre-tests were only used as a basis for creating equivalent experimental groups (by comparing the 25-, 50-, and 75-percentiles for each group). The pre-test for ABMT was also used for estimating the children's counting abilities before starting the intervention.

The digital NLET was conducted on a tablet and consisted of 15 randomized target numbers on a bounded number line from 0 to 20—similar to the virtual game described in the previous section, but without the feedback and without the narrative. The order of the tests was randomized between participants, dedicating one specific order to each researcher. The researchers also took notes on the children's behaviours during the tests—although not by any strict protocol. Such notes could be about whether the child was shy, unwilling to respond to a certain question, if they used specific strategies (pointing at objects when counting them, etc.), or something similar.

### 2.2. Participants and Procedure

As a whole, the larger study included 145 children—aged 5;9 to 6;9—from six kindergarten classes in three schools in southern Sweden, with children assigned to experimental or control conditions using convenience sampling. That is, two classes were chosen as the control, three classes were assigned to the intervention, and in one class the children were split into both groups. All schools were located in areas with medium to high socio-economic status.

After the pre-tests, a subset of the participants was chosen for the present study by stratified sampling: First, one third of the children in the experimental group was randomly assigned to the physical condition (playing the physical NLET game), excluding extreme lower- or higher-achieving participants ($N = 32$). The rest of the children in the experimental group were assigned to the virtual condition (playing the NLET game on a tablet). For the analyses, however, a corresponding subset of participants were chosen from both the virtual condition ($N = 31$) and the control group ($N = 30$) by matching participants with similar results on the pre-tests. By this stratification, we reached three comparable groups (two experimental groups and one control group) with equivalent achievement levels and number of participants.

Although the students' attendance in this case fell under the teachers' consent, the students' parents were given thorough information about the purpose and content of the study. They were also given an opportunity to decline any use of their children's data, a choice no one actually made. All data collection—physical as well as virtual—was anonymized. This procedure was approved in the ethics application [Anonymous].

### 2.2.1. Intervention Procedure and Data Gathering

The intervention took place over approximately three subsequent weeks, with one training session per week. These sessions were conducted at school, in close proximity to the children's ordinary classroom, with one researcher guiding one child. Each session started with the researcher presenting the number line, letting the child count the numbers on it, and helping out if they had difficulties. In the second and third sessions, the researchers also pointed out that number ten is in the middle, making the students aware of the possibility to use other starting points on the line to perform better.

All children's individual estimations were saved—the physical ones by storing the booklets and the virtual ones through data logs. The researchers also filled in an observation

protocol of the children's strategies (see Appendix A). For each target number, the observed strategy was categorized as (i) using the beginning as a reference point (B), (ii) using the middle as a reference point (M), (iii) using the end as a reference point (E), (iv) using proportional reasoning (Prop), or (v) using the previous number as a reference point (Prev). If the child used several reference points, all of them were noted. If nothing was observed, the protocol was left blank. The researchers also noted if the children deliberately gave the wrong response, if they pointed on the number line by mistake, etc. They also now and then posed questions (e.g., "how did you know the number was there?" or "can you tell me about your thoughts when estimating this number?") and provided feedback. In addition to the observation protocols, all training sessions were audio recorded.

In total, five researchers participated in the study— four as experimental leaders (ELs) interacting with the children and one as an additional observer. Although in most cases, each EL both guided and observed one child during play, the observer rotated amongst the ELs, assisting with completing the observation protocol. This allowed the ELs to, at least occasionally, focus on interacting with the children. All of the researchers (two doctoral students, two research assistants, and one PhD) had previous experience in the field. Before the start of the intervention, approximate guidelines for guiding the children, such as how to present the game, what questions to pose to them, how to provide/add feedback, etc., was set up. These were rehearsed during a workshop, together with rehearsal of the pre- and post-test procedures.

To ensure that the children were comfortable and happy to engage with us, the same researcher/child pair was attempted to be maintained throughout the intervention. However, we also agreed to remain flexible and responsive to each child's needs, being prepared to give the best possible support to help them recognize both their mistakes and their progress. Although the absence of a strict experimental protocol during the training sessions may limit internal validity, the children's engagement and well-being were prioritized.

*2.3. Data Analysis Plan*

Given the empirical and theoretical background presented in Section 1, and considering the young age of the participants, we expect the children in our study to demonstrate the following behaviours: (i) few will use proportional reasoning when estimating target numbers, (ii) the most common strategy will be counting forwards or backwards from a chosen reference point, and (iii) while many will show strong forward counting skills, considerably fewer will succeed in counting backwards, particularly from higher numbers.

We also consider it plausible that the children may at times ignore the upper boundary of the line. If they simultaneously apply a left-to-right counting strategy while using incorrect spacing, the resulting errors will be larger for higher target numbers. As illustrated in Figure 3, applying 50% or 150% of the correct spacing can produce an absolute error greater than 1 for target numbers of 4 and above. This justifies treating smaller target numbers as a special case. Consequently, the analyses in this paper distinguish between two target number categories: target numbers 1–4 and target numbers 5–19.

A commonly used measure in NLET studies is PAE (percentage of absolute error), which is calculated as follows: PAE = 100 × |Estimate − Target Number|/Scale of number line. In the present study, an estimation of 12 for target number 10 would then be 100 × |12 − 10|/20 = 10. However, since we in this study tone down the estimation part of the NLET—expecting the children to apply counting strategies instead—we have chosen the actual absolute error of the estimation on the NLET as an alternative measurement of accuracy. We believe this to be more appropriate and easier to interpret in the current setting. The children themselves—as well as the researchers—also often related the estima-

tions to the target numbers in units of 1 when playing the game (e.g., "That's only one step away", etc.). Since we do not compare number lines with different ranges, calculating a PAE value is unnecessary.

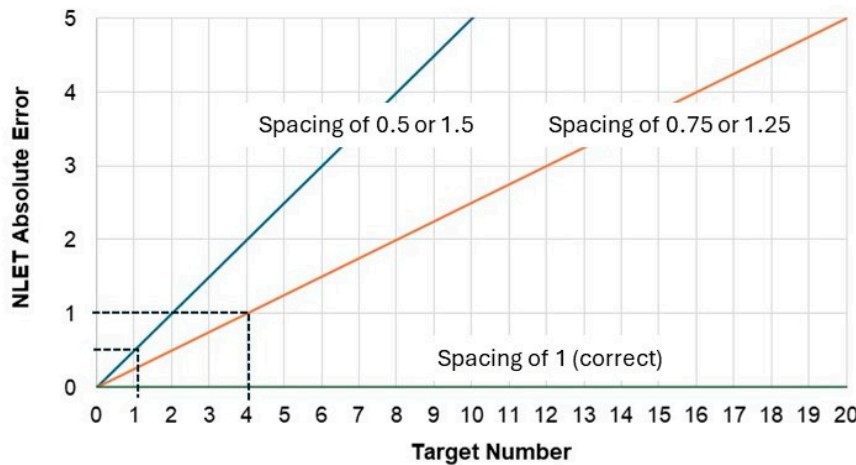

**Figure 3.** Visualization of how counting from zero with different spacing will affect NLET absolute error.

## 3. Results

In sum, 30 children in the physical condition participated in both pre- and post-tests as well as in at least two training sessions. The corresponding number of participants in the other groups was the same: $N_{virtual}$ = 30, $N_{control}$ = 30. On average, the participants in the physical condition solved between 12 and 18 tasks per session within the restricted time limit (slightly fewer in the first session, and more in the latter ones), $Mean_{TrialsPhys}$ = 17. After excluding obvious errors and mistakes from the NLET-data—all noted in the observation protocol—the children in the virtual condition solved between 24 and 30 tasks per session, $Mean_{TrialsVirt}$ = 30.

As anticipated, the physical condition was slower than the virtual one. While the average play time per training session was approximately 9 min per participant and session for the physical condition (introduction/closure excluded), it was only 6 min in the virtual condition.

### 3.1. The Children's Counting Abilities at the Start of the Intervention

The results on the ABMT pre-tests revealed that even if many of the children could count forwards way beyond 20, they also had severe difficulties counting backwards. Approximately 40% of them (all groups, *N* = 90) counted easily to 50 or above, while 76% of them managed to count to 20. However, while 71% succeeded counting backwards from 10, only 27% counted backwards correctly from 15. This finding supports the decision to treat target numbers between 1 and 4 as a special case.

### 3.2. NLET Accuracy on Pre- and Post-Tests

The first hypothesis stated that the intervention would affect the children's number line estimation abilities, measured by pre- and post-tests. To evaluate this, we set up a logistic regression model with mixed effects and repeated measures, using the following variables:

**Absolute errors from the NLET: AbsErrTest** (continuous dependent variable).

**The NLET condition: Cnd** (categorical independent variable, fixed effects). Three conditions: Physical, Virtual, and Control.

**The type of test: Test** (categorical independent variable, fixed effects). Two categories: PreTest and PostTest.

**The target number category: TargetCat** (categorical independent variable, fixed effects). Two categories: [1–4] and [5–19].

**Student subject: Id** (categorical independent variable, randomized effects).

**Interaction effects:** Three interaction effects were hypothesized: Between Test and TargetCat, between Cnd and TargetCat, and between Test and Cnd.

logit (AbsErrTest) = $\beta_0 + \left[\frac{1}{Id}\right] + \beta_1 \text{Cnd} + \beta_2 \text{Test} + \beta_3 \text{TargetCat} + \beta_4 \text{Test:TargetCat} + \beta_5 \text{Cnd:TargetCat} + \beta_6 \text{Cnd:Test}$

Given that the distribution of the NLET absolute errors deviated from a normal distribution, a generalized regression model with a gamma distribution were used in the analysis. The regression model was fitted using a step-wise–step-forward procedure, controlling for AIC–values and *p*-values from log-likelihood ratio tests, controlling for the maximum correlation of fixed effects not being larger than 0.85.

The fitting of the model resulted in the following: All the independent variables, except the interaction effect between Cnd and TargetCat, contributed significantly to the model. As hypothesized, the absolute errors decreased between pre- and post-tests, but mainly for TargetCat [5–19]. The absolute errors for TargetCat [1–4] were substantially smaller than the absolute errors for TargetCat [5–19] and remained almost stable between pre- and post-tests. The final minimal adequate for NLET absolute errors (AbsErr) performed significantly better than an intercept-only baseline model but had a weak fit: $\chi^2(7)$: 188.8, $p < 0.001$, Mc Fadden's adj $R^2 = 0.04$. We may thus conclude that Hypothesis 1 is significantly supported. The model is visualized in Figure 4, and statistical details are presented in Table 2.

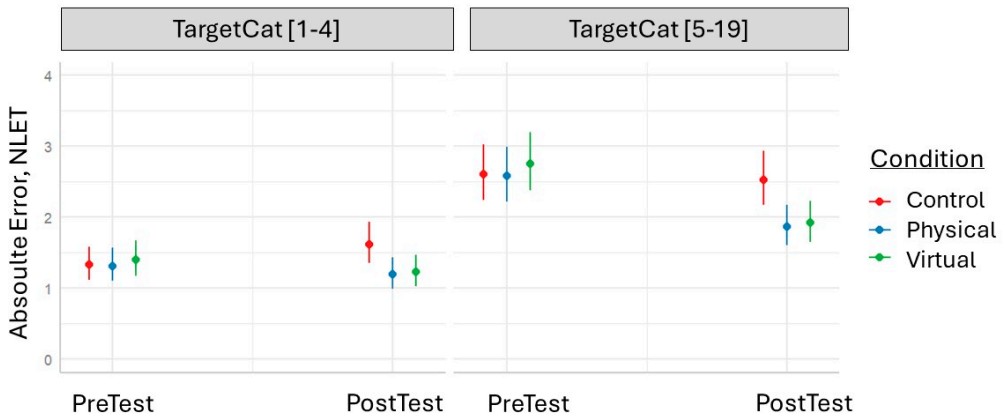

**Figure 4.** Predicted pre- and post-test NLET absolute errors for different target number categories.

**Table 2.** Summary of final minimal generalized mixed-effects regression model fitted to predict AbsErrTest.

| Random Effects | | Variance | | Std. Dev. |
|---|---|---|---|---|
| Id (*N* = 90) | | 0.11 | | 0.33 |
| **Fixed Effects** (no of obs = 2670) | **Coeff.** | **Std. Err.** | *t*-**Value** | **Pr (>|z|)** |
| Intercept | 0.28 | 0.09 | 3.00 | <0.01 ** |
| Cnd [Physical] | −0.01 | 0.11 | −0.11 | 0.92 ns |
| Cnd [Virtual] | −0.06 | 0.11 | 0.51 | 0.61 ns |
| TargetCat [5–19] | 0.68 | 0.06 | 10.71 | <0.001 *** |
| Test [PostTest] | 0.20 | 0.10 | 2.05 | <0.05 * |
| Cnd [Physical]: Test [PostTest] | −0.29 | 0.09 | −3.26 | <0.01 ** |
| Cnd [Virtual]: Test [PostTest] | −0.33 | 0.09 | −3.68 | <0.001 *** |
| TargCat [6–19]: Test [PostTest] | −0.23 | 0.09 | −2.53 | <0.05 * |

**Model statistics** AIC: 9510, Mc Fadden's adj $R^2$: 0.04, Likelihood ratio test: $\chi^2(7)$: 188.8, $p < 0.001$ ***, * $p < 0.05$, ** $p < 0.01$, *** $p < 0.001$, ns = not significant.

### 3.3. Observable Strategies During Training Sessions in Different Conditions

The second hypothesis stated that the two learning environments would influence the children's interaction patterns during the intervention in different ways. As outlined in the Method Section, the researcher systematically documented each child's NLET strategy on every trial, noting whether any—or several—of the following strategies were applied: B (beginning as a reference point), M (middle as a reference point), E (end as a reference point), Prop (proportional reasoning), Prev (previous number as a reference point), NoObs (no strategy observed). If several reference points were used, the strategy was classified as Mu (multiple reference points). The full protocol is attached in Appendix A.

To evaluate whether the strategies differed between conditions, we first examined the proportion of estimations falling into each category for each participant. If averaging these for all sessions, we can see that they differ between target number categories, but also between conditions. While the participants in the physical condition used strategy B for TargCat [1–4] for 68% of the estimations, the corresponding number for the virtual condition was 36%. However, for TargCat [5–19], the use of this strategy was almost the same in both conditions, 15% versus 14%.

For TargCat [1–4], each one of the strategies M, E, Prev, Prop, or Mu were all very uncommon in both conditions, less than 3% per category. On the other hand, for TargCat [5–19], using the midpoint as a reference point (M) was almost twice as common in the physical condition than in the virtual condition, 20% compared to 10%. For this target number category, the use of multiple reference points (Mu) was also more common in the physical condition, 7% compared to 0%, as was the use of the endpoint as a reference point, 12% in the physical condition compared to 9% in the virtual. However, as for TargCat [1–4], the strategies Prev and Prop were extremely rarely used for TargCat [5–19], less than 1% in both conditions.

Since the strategies in categories B and NoObs were the most common, and several of the other strategies—such as Prev and Prop—were rarely used, we decided to merge the categories M, E, Prop, Prev, and Mu into a single category. This decision was further motivated by the hypothesis that using any other reference point(s) than 0 would contribute to more accurate estimations for target numbers greater than 4. In addition, by merging infrequently used categories, sparsity in the data was reduced, allowing for more robust statistical analyses and enhancing power to detect meaningful effects. This merge left us with three categories: using the beginning as a reference point (B), using other reference points (MEP), and no strategy observed (NoObs).

To investigate whether the strategies changed over the course of the intervention, the proportions per training session were averaged for each condition, and the development from session 1 to session 3 was assessed using paired sample *t*-tests. The results showed that the only significant change was in the proportion of MEP strategies for TargetCat [5–19] in the physical condition: $t(27) = 4.98$, $p < 0.001$, Cohen's $d = 0.94$ (*p*-values Bonferroni-corrected using a factor of 6). Statistical details for the analyses are presented in Table 3 (physical condition) and Table 4 (virtual condition).

Independent *t*-tests comparing the proportion of different strategies between the physical and virtual condition in session 3 also revealed significant results for Strategy B, TargetCat [1–4]: $t(52) = 5.42$, $p < 0.001$, Cohen's $d = 1.48$; for Strategy NoObs, TargetCat [1–4]: $t(50) = -5.80$, $p < 0.001$, Cohen's $d = -1.58$; and for Strategy MEP, TargetCat [5–19]: $t(50) = 3.50$, $p < 0.001$, Cohen's $d = 0.96$ (*p*-values Bonferroni-corrected using a factor of 6). Thus, Hypothesis 2 is strongly supported. Statistical details for the analyses are presented in Table 5.

**Table 3.** Average proportions of used NLET strategies for physical condition, session 1 to session 3.

| | | TargetCat [1–4] | | | TargetCat [5–19] | | |
|---|---|---|---|---|---|---|---|
| **Session** | *N* | **B** | **MEP** | **NoObs** | **B** | **MEP** | **NoObs** |
| 1 | 31 | 65% | 5% | 30% | 21% | 32% | 47% |
| 2 | 30 | 64% | 4% | 32% | 16% | 42% | 42% |
| 3 | 27 | 75% | 4% | 21% | 7% | 53% | 40% |
| Paired *t*-test (sess3-sess1) Cohen's *d* | | $t(27) = 1.39$ $p = 0.18$ | $t(27) = -0.25$ $p = 0.81$ | $t(27) = -0.05$ $p = 0.27$ | $t(27) = -0.01$ $p = 0.18$ | $t(27) = 4.98$ $p < 0.001$ 0.94 | $t(27) = -0.02$ $p = 0.21$ |

**Table 4.** Average proportions of used NLET strategies for virtual condition, session 1 to session 3.

| | | TargetCat [1–4] | | | TargetCat [5–19] | | |
|---|---|---|---|---|---|---|---|
| **Session** | *N* | **B** | **MEP** | **NoObs** | **B** | **MEP** | **NoObs** |
| 1 | 30 | 28% | 4% | 67% | 13% | 18% | 68% |
| 2 | 30 | 46% | 4% | 50% | 15% | 26% | 58% |
| 3 | 25 | 33% | 2% | 65% | 14% | 31% | 54% |
| Paired *t*-test (sess3-sess1) | | $t(25) = 0.76$ $p = 0.45$ | $t(25) = -1.29$ $p = 0.21$ | $t(25) = -0.26$ $p = 0.80$ | $t(25) = -0.10$ $p = 0.93$ | $t(25) = 2.61$ $p = 0.02$ | $t(25) = -2.18$ $p = 0.04$ |

**Table 5.** Comparison between proportions of used NLET strategies in session 3 for two different conditions.

| | | TargetCat [1–4] | | | TargetCat [5–19] | | |
|---|---|---|---|---|---|---|---|
| **Session 3** | *N* = 50 | **B** | **MEP** | **NoObs** | **B** | **MEP** | **NoObs** |
| Welch two-sample *t*-test (physical-virtual) Cohen's *d* | | $t(52) = 5.42$ $p < 0.001$ 1.48 | $t(36) = 0.57$ $p = 0.57$ | $t(50) = -5.80$ $p < 0.01$ −1.58 | $t(47) = -2.19$ $p = 0.03$ | $t(50) = 3.58$ $p < 0.001$ 0.96 | $t(30) = -1.61$ $p = 0.12$ |

*3.4. The Effect of NLET Strategies on NLET Performance During Training Sessions*

The third hypothesis stated that the utilized NLET strategies would influence the performance on the NLET during play. To investigate whether this was the case, we set up a similar logistic regression model with mixed effects and repeated measures as in Section 3.2, using the following variables:

**Absolute Error from the NLET during Sessions: AbsErrSess** (continuous dependent variable).

**Strategy** (categorical dependent variable): Three categories: B, MEP, and NoObs.

**The NLET condition: Cnd** (categorical independent variable, fixed effects). Two conditions: Physical and Virtual.

**Training Session: Session** (categorical independent variable, fixed effects). Three categories: Session1, Session2 and Session3.

**The target number category: TargetCat** (categorical independent variable, fixed effects). Two categories: [1–4] and [5–19].

**Student subject: Id** (categorical independent variable, randomized effects).

**Interaction effects:** Three interaction effects were hypothesized: Between Cnd and TargetCat, between Session and TargetCat, and between Strategy and TargetCat.

logit (AbsErrSess) = $\beta_0 + \left[\frac{1}{Id}\right] + \beta_1$Cnd + $\beta_2$Session + $\beta_3$TargetCat + $\beta_4$Strategy + $\beta_5$Cnd:TargetCat + $\beta_6$Session:TargetCat + $\beta_7$Strategy:TargetCat

As in Section 3.2, a generalized regression model with a gamma distribution were used in the analysis. The model was fitted by a step-wise–step-forward procedure, controlling for

AIC values and *p*-values from log-likelihood ratio tests, and controlling for the maximum correlation of fixed effects not being larger than 0.85. The fitting of the model resulted in the following: only the variables Session, TargetCat, and Strategy contributed significantly to the model, and the only significant interaction effect was the one between Strategy and TargetCat. As expected, the absolute errors decreased gradually between training sessions, but mainly for TargetCat [5–19]. As in Section 3.2, the absolute errors for TargetCat [1–4] were substantially smaller than the absolute errors for TargetCat [5–19] and remained almost stable between pre- and post-tests.

The final minimal adequate model for NLET absolute errors during sessions (AbsErrSess) performed significantly better than an intercept-only baseline model and had a satisfactory fit: $\chi^2(7)$: 537.7, $p < 0.001$, Mc Fadden's adj $R^2 = 0.13$. Thus, the findings indicate that the strategies significantly influenced NLET performance, providing support for Hypothesis 3, even though the experimental condition (virtual versus physical) did not directly influence the accuracy of the children's estimations. The model is visualized in Figure 5, and statistical details are presented in Table 6.

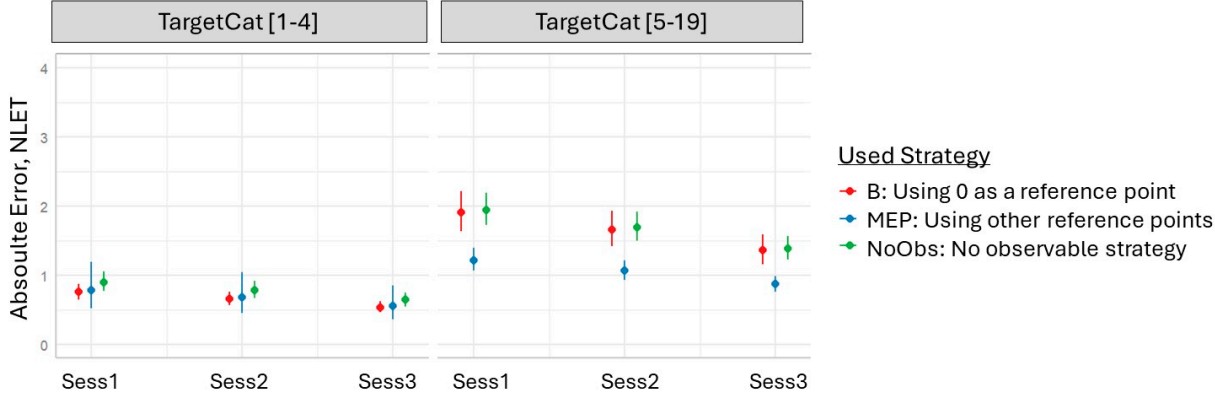

**Figure 5.** Predicted NLET absolute errors during training sessions in relation to utilized strategies.

**Table 6.** Summary of final minimal generalized mixed-effects regression model fitted to predict AbsErrSess.

| Random Effects | | Variance | | Std. Dev. |
|---|---|---|---|---|
| Id (*N* = 61) | | 0.15 | | 0.39 |
| **Fixed Effects** (no of obs = 4091) | **Coeff.** | **Std. Err.** | *t*-value | **Pr (>\|z\|)** |
| Intercept | −0.29 | 0.07 | −4.78 | <0.001 ** |
| Session [2] | −0.14 | 0.04 | −3.71 | <0.001 ** |
| Session [3] | −0.34 | 0.04 | −8.75 | <0.001 ** |
| TargetCat [5–19] | 0.93 | 0.07 | 14.06 | <0.001 *** |
| Strategy [MEP] | 0.04 | 0.19 | 0.21 | 0.83 ns. |
| Strategy [NoObs] | 0.18 | 0.07 | 2.62 | <0.01 ** |
| TargetCat [5–19]: Strategy [MEP] | −0.49 | 0.20 | −2.47 | <0.05 * |
| TargetCat [5–19]: Strategy [NoObs] | −0.16 | 0.08 | −1.87 | 0.06. |

**Model statistics** AIC: 9975, Mc Fadden's adj $R^2$: 0.13, Likelihood ratio test: $\chi^2(7)$: 537.7, $p < 0.001$ ***, * $p < 0.05$, ** $p < 0.01$, *** $p < 0.001$, ns = not significant.

### 3.5. The Effect of the Used Strategies on Post-Test Performance

The fourth hypothesis stated that the learned strategies, as well as the condition (virtual/physical), would influence the children's NLET performance on the post-test. To evaluate this, we set up a generalized regression model with the mean value of the absolute errors on the post-test (AbsErrMean) as a dependent variable, focusing on target numbers

between 5 and 19—this was decided since the former analyses strongly indicate that the estimations for smaller target numbers showed only minimal improvement between pre- and post-tests. We then predicted this measure to depend on the condition, as well as the child's use of MEP strategies during the last training session (session 2 or session 3). Since the number of training trials differed between conditions, as well as between participants, we also wanted to control for this in the analysis. Thus, the following variables were used to formulate the model:

**Mean value for the absolute errors for TargCat** [5–19] **on the post-test: AbsErrMean** (continuous dependent variable, one value per participant).

**The NLET condition: Cnd** (categorical independent variable). Two conditions: Physical and Virtual.

**Proportion of MEP for TargCat** [5–19] **during the latest training session: StratMEP519** (continuous dependent variable, one value per participant).

**Total number of training trials for TargCat** [5–19]: **NumTrials519** (continuous independent variable, one value per participant).

**Interaction effects:** Three interaction effects were hypothesized: Between Cnd and StratMEP519, between Cnd and NumTrials519, and between StratMEP519 and NumTrials519.

$$\text{logit (AbsErrTestMean)} = \beta_0 + \beta_1\text{Cnd} + \beta_2\text{StratMEP519} + \beta_3\text{NumTrials519} + \beta_4\text{Cnd:StratMEP519} + \beta_5\text{Cnd: NumTrials519} + \beta_6\text{StratMEP519:NumTrials519}$$

As before, a generalized regression model with a gamma distribution were used in the analysis. The model was fitted by a step-wise–step-forward procedure, controlling for AIC values and *p*-values from log-likelihood ratio tests, and controlling for the maximum correlation of fixed effects not being larger than 0.85. The fitting of the model resulted in the following: only the variables StratMEP519 and NumTrials519 contributed significantly to the model, and no interaction effects occurred. The final minimal adequate model for AbsErrMean performed significantly better than an intercept-only baseline model but had a modest fit: $\chi^2(2)$: 14.0, $p < 0.001$, Mc Fadden's $R^2 = 0.08$. Thus, the findings indicate that the strategies used in the latest training session, together with the number of trials, were the only factors significantly influencing NLET performance on the post-test. Consequently, Hypothesis 4 was only partially supported. The model is visualized in Figure 6, and statistical details are presented in Table 7.

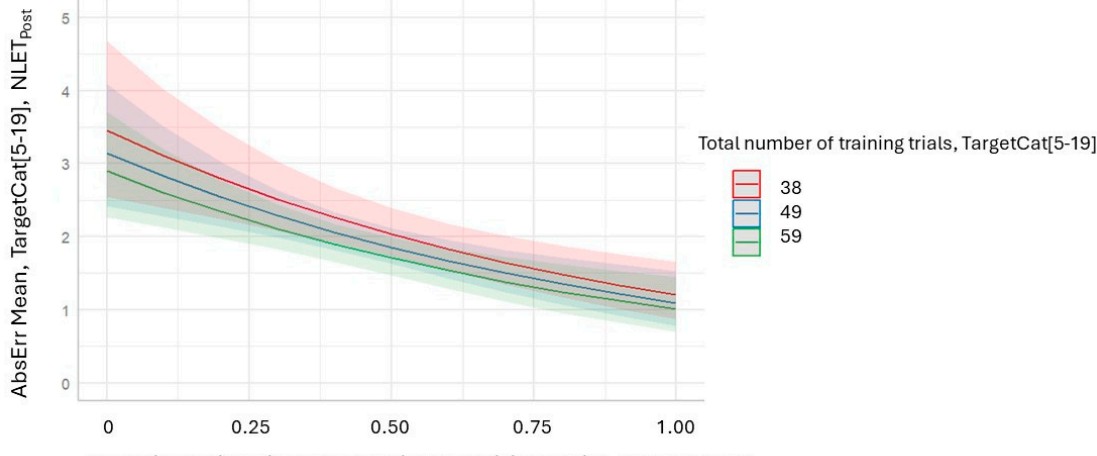

**Figure 6.** Predicted NLET absolute errors on post-test in relation to MEP strategies and amount of training.

**Table 7.** Summary of final minimal generalized regression model fitted to predict AbsErrMean.

| (*N* = 60) | Coeff. | Std. Err. | *t*-Value | Pr (>\|z\|) |
|---|---|---|---|---|
| Intercept | 1.56 | 0.27 | 5.71 | <0.001 *** |
| StratMEP519 | −1.06 | 0.27 | −3.91 | <0.001 *** |
| NumTrials519 | −0.01 | 0.004 | −2.05 | <0.05 * |

**Model statistics** AIC: 160, Mc Fadden's $R^2$: 0.08, Likelihood ratio test: $\chi^2$(2): 14.0, $p < 0.001$ ***, * $p < 0.05$, *** $p < 0.001$.

### 3.6. Qualitative Findings

As the intervention progressed, it became increasingly clear how different the two learning environments actually were—not only in terms of the children's behaviour, but also in the interaction between researchers and children. Consequently, to gain a deeper understanding of the activities during the intervention, and to examine whether and how the learning situation as a whole was influenced by the learning material used, quantitative data was complemented with qualitative findings. By analyzing audio files from the training sessions and by exploring data logs and time stamps for different activities, we can conclude that the conditions differed in more ways than with respect to the physical properties of the learning material itself. More specifically, the learning environment strongly affected the talk between the child and the researcher, but it also influenced the child's eagerness to respond and to make an effort before making the actual estimations.

In the virtual condition, the children were almost always a bit excited and competitive during play. They responded quickly but were also sometimes negatively affected by the feedback—especially by negative—and occasionally they ignored it or even clicked it away. When playing the virtual game, some of the lower-achieving children also often made wild guesses, not really making an effort. Since it was possible to make mistakes in the virtual condition by accidentally touching the screen or by unintendedly lifting the finger from the number line at the wrong location, the children were sometimes frustrated. Nevertheless, most of the children learned the interface very fast, being really eager to hit the target number on the spot.

In the physical condition, lower-achieving children were instead more reluctant to respond, sometimes playing around with the animals, drawing figures on the paper sheets, or scribbling. Many children also talked a lot in the physical condition, occasionally went off task, and had to be told to stay focused. On the other hand, since the researcher in this case could adapt the feedback to each participant and situation, it was possible to ensure that the child really listened to it and understood it. Interestingly, the children in the physical condition also often mitigated negative feedback, saying things like, "Well, the line I drew points towards the target number. . .", or "He (the frog) could just stretch out his hand to reach it (the food). . .". In the virtual condition, on the other hand, the researcher had a very limited time to complement the game's feedback before a new target number was presented. The children also had a limited time to engage in the feedback, and they rarely elaborated on the accuracy of their estimations—even if they could say things like, "Oh, that wasn't good!" or "Spot on!". Although the possibility to add verbal feedback was limited in the virtual condition, the researchers often slipped in comments about the accuracy of the estimations, either by reinforcing positive feedback with expressions like, "Great work, you saw that was close to X!", by mitigating negative feedback by saying, "Oh, but that was close to Y!", or something similar.

If roughly estimating the time spent on different activities during one NLET (i.e., the average number of seconds spent on one single estimation of a target number), the difference between conditions becomes evident. As shown in Figure 7, the extra time in the physical condition was not only spent on practicalities and small talk, but also on solving

the NLET (response time = 9 s in the physical condition compared to 5 s in the virtual condition) and on feedback conversations (10 s in the physical condition compared to 6 s in the virtual condition).

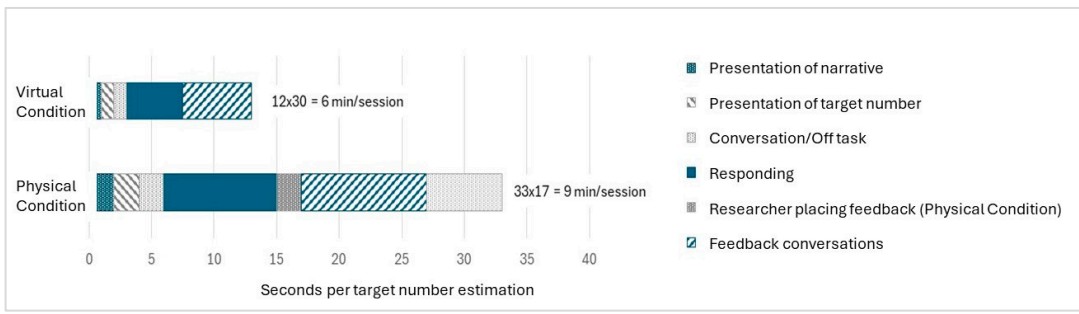

**Figure 7.** Estimated time for different activities during game play in two conditions (interpreted from audio files from session 2).

In sum, one of the most telling qualitative findings is how the verbal interaction between researcher and child differed between conditions. Examples of the conversations (two from each condition) are presented in Appendices B–E.

*3.7. Summary*

The purpose with the present study was to evaluate whether two different types of an NLET training material—a pen- and paper-based game versus a virtual game—would affect children's learning in different ways, influencing behaviours and strategies as well as NLET performance on a post-test. We also hypothesized that the utilized strategies—such as using multiple reference points along the number line—would be tightly linked to the accuracy on the numerical estimations. However, we also proposed that these strategies would not automatically transfer between learning environments (physical/virtual). This was investigated by letting the children that played the physical game conduct the same post-test as the children playing the virtual game—i.e., on a tablet.

The results reveal that children in both experimental conditions improved their NLET performance on the post-test compared to a control group. It was also shown that the children's strategies during the intervention differed between conditions. While the children in the physical condition significantly increased their use of other reference points than zero for larger target numbers (Strategy [MEP], TargetCat [5–19]) between training sessions 1 and 3, no such development was shown for the children in the virtual condition. When comparing the use of such strategies during the final training session between conditions, these strategies were also more frequent amongst the children playing the physical game. In turn, the use of this strategy had a significant impact on the accuracy of the estimations for larger target numbers (TargetCat [5–19]). A frequent use of Strategy [MEP] for TargetCat [5–19] in the final training session was also a significant predictor for the performance on the digital post-test—together with the total number of training trials for this target number category. Still, the type of learning material (physical/virtual) did not directly affect the performance on the NLET—neither during training sessions nor on the post-test. Instead, this was over-shadowed by the other variables.

When complementing quantitative results with qualitative data, we can conclude that a transformation from a physical NLET game to a virtual counterpart will change the learning environment in a multitude of ways—not only speeding up the problem-solving process but also decreasing valuable math-talk between researcher and child.

## 4. Discussion

The finding that the observed strategies significantly predicted NLET performance is in line with Petitto et al.'s study (1990), stating that the capacity to correctly estimate target numbers on the number line is primarily related to the participants' ability to utilize the line's spatial properties together with a range of reference points (i.e., the midpoint, the endpoint, or a combination of these and the starting point) to calibrate their estimations (see also H. C. Barth & Paladino, 2011; Schneider et al., 2008). The finding that the estimations of smaller target numbers were substantially more accurate—independently of the strategies used as well as the intervention as a whole—is also in line with some of the findings in the study by White and Szűcs (2012). The absence of performance differences between conditions is consistent with the findings of Piatt et al. (2016), who evaluated virtual and physical NLET assessment to be equivalent. However, Piatt and colleagues focused solely on students' errors on the NLET, without examining their strategies or behaviours. Moreover, their material is not fully comparable to the games used in our study.

Consequently, the most interesting result from our study is the effect of condition on observable behaviours. The pen and paper game apparently afforded the children more expressive interaction, making them utilize reference points along the number line more often, but also increasing their talk about their estimations during play. And even if this condition was more time-consuming, reducing the number of trials per training session, these qualities seemed to compensate for the loss of practice opportunities. In the physical condition, the researcher also had more time to elaborate on feedback, ensuring that the child paid attention to it and understood it. The observed differences between conditions, both in terms of children's interaction with the game and their interaction with the researchers, suggest that the physical and virtual versions of the game provided children with two distinct learning situations—leading to different pathways towards improved NLET performance. This topic is further examined in the passage below.

### 4.1. Two Different Pathways Towards Improved NLET Performance

If further exploring the influence of how the two variables Strategy [MEP] and the number of training trials affected the accuracy on the NLET for target numbers between 5 and 19, we can see that, to achieve a good result, a participant could either perform many trials or use MEP strategies more frequently. As an example, having estimated 38 target numbers during the intervention and using MEP strategies for 51% of the trials during the last training session, gives a predicted value for AbsErrMean of 2.0 (Figure 8). Another option would be to perform 59 corresponding trials but to only use MEP strategies for 35% of the trials during the last session (Figure 8). Since the first example corresponds well to a participant in the physical condition and the second to a participant in the virtual condition, we can conclude that, even if the post-test performance was equivalent in the two groups, the participants did not perform the same activities during training.

If complementing the quantitative results with qualitative data, we may suggest that the two learning materials led to two different learning experiences. While the physical game was substantially slower (sometimes due to off-task activities), it also led to richer interactions and a had a higher quality per training trial. The child not only attended more to the feedback, sometimes reflecting on it in a constructive way, but the conversations with the researcher were also more elaborated. In the virtual condition, the training was, on the other hand, quicker and more mechanistic. These children sometimes hurried through the exercises, neglected feedback, and had a competitive attitude.

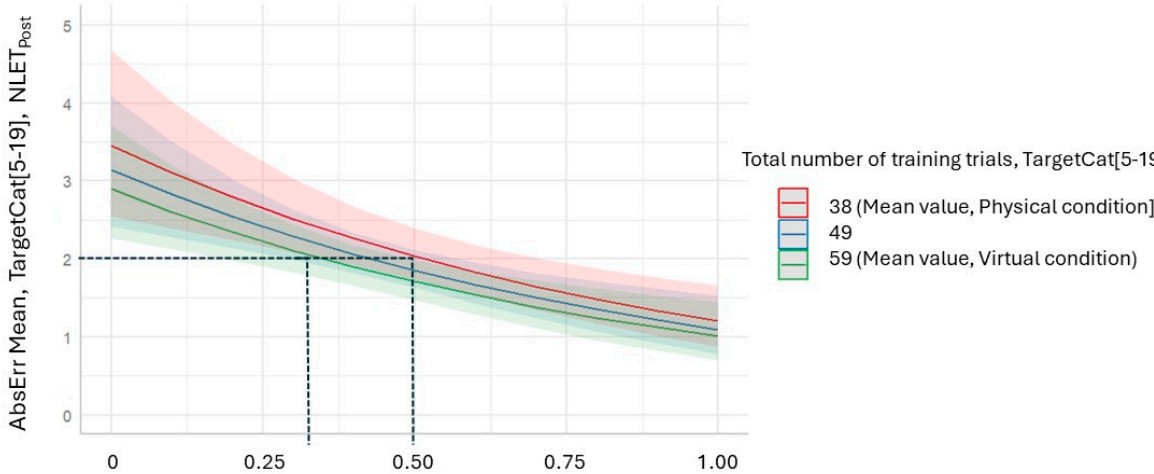

**Figure 8.** Predicted NLET absolute errors on post-test for two different set of values—corresponding to average player in each condition (virtual/physical).

We could sum this up as follows: In the physical condition, the child's activities were in the centre, with the researcher steering and controlling the game, and with the learning material as a useful tool between the two., In the virtual condition, however, it was the game itself that commanded the attention of both the child and the researcher, guiding their focus and actions, and sometimes actually hindering the conversation between researcher and child. That touchscreen technology and EdTech games can have this effect on child–adult interaction has been shown in earlier studies (de Vries et al., 2021; Carr & Dempster, 2021; Hiniker et al., 2018). Physical math games do apparently lead to more constructive math-talk during play compared to their digital counterparts. This effect is visualized in Figure 9.

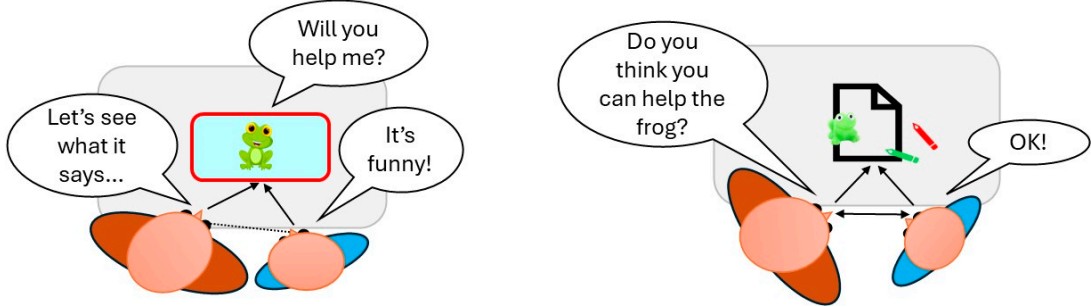

**Figure 9.** Researcher–child–game interaction, virtual condition (**left**), physical condition (**right**).

### 4.2. Limitations and Suggestions for Future Research

Even though we believe the present work to be unique, the study of course has limitations. First of all, converting a physical learning material into a digital learning tool (or vice versa) almost always alters the learning situation in more than one way, lowering internal validity when making comparisons (see also M. A. Rau, 2020). The present study is no exception. It is thereby hard to determine the exact features (the physical/virtual interface, the provided feedback, etc.) in each learning situation that had the largest effect. To clarify this, more research is needed—preferably by balancing the researcher's control over the learning environment and making multiple overlapping comparisons. The study would also have benefitted from a more rigorous experimental design, with a more randomized setup, and with an active control group participating in other activities

together with the researchers. This would have excluded the possibility of the post-test results being affected by the children's general active engagement (i.e., the test-effect) and/or by their habituation to the researchers.

Moreover, the trustworthiness of the observational data could have been enhanced through coding conducted by multiple researchers. The study would also have benefitted from a proper observation protocol during the NLETs in the pre- and post-tests—similar to the one used during training sessions. Unfortunately, the gathered notes during these tests were not detailed enough for evaluating the children's NLET strategies. This makes it impossible to be certain of whether the children kept (or remembered) the learned strategies when not being scaffolded and guided by the researcher—as in the training sessions.

Of course, the one-to-one training sessions set up in the present study are also hard to transfer to a real learning situation in an ordinary school, where resources are scarce and teachers have a hard time giving each child individual support and guidance. In this case, a virtual number line game would be preferable—especially if providing elaborate and encouraging feedback. However, a substantial body of research has shown that children working individually with EdTech games may very easily ignore feedback all together (Meyer et al., 2025; Tärning et al., 2020) or fall into unfruitful wheel-spinning behaviours (Beck & Gong, 2013). Thus, it is very important to carefully monitor game-playing students' behaviour.

Another option could be to practice the NLET in a larger group. The teacher could, for example, show an empty number line together with a target number on a white-board, and let the children—one at a time—come forward to the board and make a mark. If using a digital interface, the correct position of the target number could then be accurately presented, making it possible to discuss the number's position—as well as the estimation—with the group as a whole. In fact, we have shared such a material with the teachers in the schools participating in this study—with positive reactions. It would be very interesting to study this type of learning situation and compare it with individual practice.

Finally, since many children in our study were not only high-performing but also behaving very well, one might hypothesize that an equivalent study with more challenging and lower-performing children could have gained even larger differences between conditions. This is because teacher support and student monitoring are even more important in such learning environments. However, the schools participating in our study were situated in middle- to high-socio-economic areas. Thus, the results may not generalize to other contexts.

**Author Contributions:** Conceptualization, E.-M.T. and M.M.R.; Methodology, E.-M.T., M.M.R., and S.H.; Formal Analysis, E.-M.T. and M.M.R.; Investigation, E.-M.T., M.M.R., and S.H.; Writing—Original Draft, E.-M.T. and M.M.R.; Writing—Review and Editing, S.H. All authors have read and agreed to the published version of the manuscript.

**Funding:** This research received no external funding.

**Institutional Review Board Statement:** The study was conducted in accordance with the Declaration of Helsinki and approved by Swedish Ethical Review Authority (protocol code 2023-03043-01, 2023-06-19).

**Informed Consent Statement:** Informed consent was obtained from all subjects involved in the study.

**Data Availability Statement:** The raw data supporting the conclusions of this article will be made available by the authors on request.

**Conflicts of Interest:** The authors declare no conflict of interest.

## Appendix A. The Observation Protocol

| Number | Target No | Starting point | | | | Scaling | | Comments |
|---|---|---|---|---|---|---|---|---|
| | | Beginning | End | Middle | *Other* | "Correct" Linear | "Incorrect" Logaritmic | |
| Frog | Try to ask the child a question about their thoughts/strategies | | | | | | | |
| Frog | 1 | | | | | | | |
| Frog | 10 | | | | | | | |
| Frog | 2 | | | | | | | |
| Frog | 5 | | | | | | | |
| Frog | 15 | | | | | | | |
| Frog | 14 | | | | | | | |
| Frog | Try to ask the child a question about their thoughts/strategies | | | | | | | |
| Frog | 19 | | | | | | | |
| Frog | 11 | | | | | | | |
| Frog | 8 | | | | | | | |
| Frog | 17 | | | | | | | |
| Kangaroo | 10 | | | | | | | |
| Kangaroo | Try to ask the child a question about their thoughts/strategies | | | | | | | |
| Kangaroo | 3 | | | | | | | |
| Kangaroo | 4 | | | | | | | |
| Kangaroo | 19 | | | | | | | |
| Kangaroo | Try to ask the child a question about their thoughts/strategies | | | | | | | |
| Kangaroo | 18 | | | | | | | |
| Kangaroo | 12 | | | | | | | |
| Kangaroo | 13 | | | | | | | |
| Kangaroo | 5 | | | | | | | |
| Kangaroo | 6 | | | | | | | |
| Kangaroo | 7 | | | | | | | |
| Rabbit | 10 | | | | | | | |
| Rabbit | Try to ask the child a question about their thoughts/strategies | | | | | | | |
| Rabbit | 9 | | | | | | | |
| Rabbit | 8 | | | | | | | |
| Rabbit | 1 | | | | | | | |
| Rabbit | 5 | | | | | | | |
| Rabbit | 4 | | | | | | | |
| Rabbit | Try to ask the child a question about their thoughts/strategies | | | | | | | |
| Rabbit | 15 | | | | | | | |
| Rabbit | 16 | | | | | | | |
| Rabbit | 17 | | | | | | | |
| Rabbit | 18 | | | | | | | |

## Appendix B. Conversations

**Participant 1005 (virtual condition, session 2)**
Game: Hello, I'm George the frog.
Participant [Giggling]
Game: I'm out and about, looking for food. I was told there should be some food somewhere along this line. Could you help me find it, please?

Participant [Giggling]

Game: Number 1

Researcher: OK, where do we have number 1?

Participant: There...

Researcher: Nice, great!

Game: 1. Oh, spaghetti, mmmmm!

Game: Number 10.

Participant: Oh, that's easy [clicks in the middle]

Researcher: Yes, that's in the middle.

Game: 10. A sandwich, that was nice!

Researcher: Really good, super!

Game: Number 2.

Game: 2. Cinnamon bun. Yum!

Researcher: Great.

Participant [Distracted]

Researcher: OK, should we focus on doing this now?

Participant: Mmm...

Game: Number 5.

Participant [Counts silently from 0 but takes too large steps]

Researcher: So, you're counting?

Participant: OK... yes...

Game: 5. The food was over there.

Researcher: Try taking smaller steps, and it might work out...

Participant: Mmm

Researcher: Mmmm

Game: Number 15.

Researcher: OK, where do we have number 15?

Participant [Counts silently from 20 and downwards but takes too large steps and ends up at 13]

Researcher: Yes, it was so close, it was up there as well! [refers to the part of the number line between 10 and 20]

[Feedback clicked away]

Game: Number 14.

Participant [Clicks directly]

Game: 14. The food was over there.

Researcher: Oh, but that was close.

Participant: No, that was far away!

Researcher: No, it wasn't far away at all!

## Appendix C. Conversations

**Participant 6007 (virtual condition, session 3)**

Game: Number 15.

Participant: Number 15 [counts silently]. Can it be there?

Game: 15. Oh, a hot dog! Nice!

Researcher [simultaneously with the game]: Well done! And you can count either from 10, or from zero as you did there, or maybe down from 20.

Game: Number 14

Researcher: Number 14 then?

Participant: Number 14 is there. There!

Researcher: Wow, how did you know that?

Game: 14. Oh, a hamburger, yum!

Participant [simultaneously with the game]: Because there is 15, and then I just need to take one step down, and then it's there. . .

Game: Number 19

Participant: 19 is there.

Researcher: Yes, exactly, and then you can think like, there's 20, and then it is only one step down. . .

Game: 19. Oh, a pizza, thanks!

Participant [simultaneously with the game]: I know where 19 is, because I know where the pizza is [referring to previous tasks—incorrectly though, since the type of food is randomly selected in the virtual condition]

Researcher [Laughs]

Game: Number 11.

Participant: It will be the farthest away [perhaps still referring to 19].

Researcher: Well, I hope so. . .

Participant: Number 9.

Researcher: No, 11! It was 11!

Participant: 11. . . 11 OK, then it will be there. . .

Researcher: Well, nice!

Game: 11. Spaghetti. Mmmm!

Participant [imitates game]: Mmmm!

Researcher [also imitating]: Mmm.

Game: Number 8.

Participant [counts silently]: It might be there, perhaps?

Researcher: Oh, 8 was up there. . .

Game: Hmm, the food was over there.

Researcher [simultaneously with the game]: Shall we try counting down from 10 next time? 10, 9, 8, maybe that will make it easier to find?

Game: Number 17.

Researcher: 17.

Participant: 17 must be there!

Game: 17.

Researcher [simultaneously with the game]: Yes, that's totally correct. But how did you know that?

Game: A milkshake, wonderful!

Participant [simultaneously with the game]: Just because if you take 20, and then go down. . .

Researcher: Yes, and then you count down. . .

## Appendix D. Conversations

**Participant 1012, (physical condition, session 2)**

Researcher: If we take number 11, where do you think that is?

Participant: 11. Easy!

Researcher: Is that also easy?

Participant [uses 10 as a reference point and draws a line]

Researcher: Here comes another hot dog.

Researcher [presents the feedback]: That was really. . . wow, look! [the estimation was very accurate]

Participant: Before, I did this wrong. I did it on one [referring to one step from 10] but it is a bit further away, so now I measured a bit further, and then it was exactly on the spot.

Researcher: Precisely, right? You know exactly how to think.

Researcher: Now, number 8, where do you think that is?

Participant [draws a line without any specific strategy]

Researcher: There?

Researcher [presents the feedback]: That was a little bit further away, wasn't it? Number 8 was there. . .

Participant: It still points at the pizza [refers to the line that is a bit crooked]

Researcher: You mean this? [points at the line]

Participant: Yes

Researcher: Does this point at the pizza?

Participant: Yes

Researcher: OK, it points a little bit at the pizza.

Participant: He [the frog] can see the pizza

Researcher: Yes, he sees the pizza, definitely.

## Appendix E. Conversations

**Participant 6012 (physical condition, session 2)**

Researcher [presents target number]

Participant: 11! [starts counting and pointing from 0 and prepares to draw a line but stops] No! Here is ten. . . och here is eleven! [Draws line].

Researcher: Well, you realized that! That was great!

Participant: It won't be correct.

Researcher: Oh, but it will, Look! [really good estimation]

Participant: Ohh [happy]

Researcher: There it is! It was great to think like that, wasn't it? Ten is in the middle and then we just take a step that way [points at the number line]. Really good.

[Participant changes animal]

Researcher: Number 8 then, where do you think that is?

Participant [guesses] Here! No. . . [starts counting and pointing from 0] There [draws a line]. It's wrong.

Researcher: Do you want to change it?

Participant: No

Researcher [presents the feedback]: But you see, it's just one tiny leap away.

Participant: Just a tiny leap.

Researcher: A tiny tiny leap.

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
