# Peer review of "Virtual Versus Physical Number Line Training for 6-Year-Olds: Similar Learning Outcomes, Different Pathways"

_education, doi:10.3390/educsci15101350_

Round 1
Reviewer 1 Report
Comments and Suggestions for Authors
Thank you for conducting and sharing this interesting research study. The manuscript is clearly written and organized and includes current research papers in the references. As much of the teaching and learning today has moved into the digital space, this study is important in helping us understand the differences in children’s use of strategies to solve NLET.
One potential evidence that supports why your results may have indicated that children who participated in the physical (pencil/paper) teaching and learning experience were able to develop more strategies related to solving NLET could include research from neurosciences. These research have found that writing activities a much broader network of neural circuits in the brain than engaging digitally (e.g., typing). While not specific to math learning, including some of these literature may strengthen the conclusions of your study.
The manuscript is scientifically sound. However, I’m concerned about the lack of protocol in facilitating the interactions as this may mean that different children may have been scaffolded in their learning inconsistently. It would also be important to share who the research team is. Were they trained to interact with the children so that there’s some reliability/fidelity in how they facilitate the interactions? It might therefore be difficult to reproduce this study if there isn’t a protocol for how the interactions /interventions are facilitated or how the research team is trained to ensure there is a consistency in how children’s learning is scaffolded or how feedback is provided.
Author Response
Reply, reviewer 1:
Comment: One potential evidence that supports why your results may have indicated that children who participated in the physical (pencil/paper) teaching and learning experience were able to develop more strategies related to solving NLET could include research from neurosciences. This research has found that writing activates a much broader network of neural circuits in the brain than engaging digitally (e.g., typing). While not specific to math learning, including some of this literature may strengthen the conclusions of your study.
Reply: The neurocognitive differences between physical and digital interaction are really interesting! And, as you point out, multimodal neural traces of pen- and paper-activities are both stronger and richer than digital screen-based interactions – which may improve memorization etc. Still, we do feel that this is perhaps slightly out of the scope of this study – since we focus on how children engage with the material during the actual problem-solving. With an embodied/embedded perspective on cognition, this means putting the learning material in the centre of the analysis, stating that the properties of the material may scaffold and mediate the teaching/learning process. If the results would have proven that the children in the physical condition had significantly stronger learning gains than the children in the virtual condition, this might have been of interest to point out in the discussion. However, this was not the case. Nevertheless, future studies with a stronger focus on how different types of multimodal interaction may scaffold learning of, for example, numbers or arithmetic operations, by constructing broader neurocognitive traces of the symbols etc., would here be of great interest.
Comment: The manuscript is scientifically sound. However, I’m concerned about the lack of protocol in facilitating the interactions as this may mean that different children may have been scaffolded in their learning inconsistently. It would also be important to share who the research team is. Were they trained to interact with the children so that there’s some reliability/fidelity in how they facilitate the interactions? It might therefore be difficult to reproduce this study if there isn’t a protocol for how the interactions /interventions are facilitated or how the research team is trained to ensure there is a consistency in how children’s learning is scaffolded or how feedback is provided.
Reply: New information has been added at row 373-386 in the manuscript. It should be noted, however, that even if we started out with a clear idea on how to guide the children and what to say to them, the learning material (physical/virtual) turned out to heavily affect the researcher/child interaction during play – which is also one of the results. See also row 614-618, 685-688, 703-713, for clarification. Changes also made in row 295-296.
Reviewer 2 Report
Comments and Suggestions for Authors
The manuscript investigates whether six-year-old children’s number line estimation abilities differ when trained using physical (pen-and-paper) or virtual (tablet-based) materials. Using a mixed-methods design, the study combines pre-/post-tests, observational data on strategies, and qualitative analysis of child–researcher interactions. Results show that while both conditions improved children’s performance, the pathways differed: physical training fostered more advanced strategy use and richer math talk, whereas virtual training allowed more practice trials. The paper is timely, relevant, and well written, with valuable implications for early numeracy education and the design of digital learning tools.
The study addresses an important question in early mathematics education, situated within debates on digital versus physical learning tools. The integration of quantitative and qualitative data is a strength, providing a more holistic understanding of learning processes. The paper is well grounded in the literature on number line estimation and embodied cognition. However, the internal validity of the comparison is weakened by differences in task structure (time limits, trial numbers, researcher-led vs. automated feedback). The contribution would be strengthened by clarifying novelty relative to prior studies, more critical treatment of methodological confounds, and tighter integration of the qualitative findings with the quantitative results.
The hypotheses are clearly stated (H1–H4), but some are only partially testable due to methodological differences between conditions. For example, H2 (learning environment affects strategies) is well supported, but disentangling whether differences are due to medium (physical/virtual) or to task design (feedback type, number of trials) is challenging. This limitation should be emphasized.
The use of stratified sampling and mixed-effects models is appropriate. However, collapsing multiple strategies (midpoint, endpoint, proportional reasoning, previous-number) into one “MEP” category reduces analytical granularity. The rationale should be explained more thoroughly, and alternative analyses considered.
Inter-rater reliability for coding strategies is not reported. If coding was done by multiple raters, please report reliability; if by a single rater, acknowledge this as a limitation.
The control group did not engage in any alternative math activity. This makes it difficult to isolate the effects of number line training versus simple exposure to researcher attention. A stronger design would include an active control. This limitation should be addressed in the Discussion.
The study context (Swedish kindergartens, medium–high SES) should be more explicitly discussed in terms of cultural and demographic constraints. Results may not generalize to other contexts.
The one-to-one researcher-led format limits transferability to typical classroom practice. Future directions could explore group-based adaptations.
The qualitative findings (e.g., richer math talk in the physical condition) are insightful but somewhat separated from the main argument. Stronger integration with quantitative results would enrich the contribution.
Author Response
Reply, reviewer 2:
Comment: The contribution would be strengthened by clarifying novelty relative to prior studies,
Reply: Added a section to motivate the study further, row 208-218.
Comment: The hypotheses are clearly stated (H1–H4), but some are only partially testable due to methodological differences between conditions. For example, H2 (learning environment affects strategies) is well supported, but disentangling whether differences are due to medium (physical/virtual) or to task design (feedback type, number of trials) is challenging. This limitation should be emphasized.
Reply: Really good point! This is one of the great challenges with comparing conventional/physical and virtual learning environments. This has been further clarified in row 240-249. Have also changed row 225-226 for clarification.
Comment: …the internal validity of the comparison is weakened by differences in task structure (time limits, trial numbers, researcher-led vs. automated feedback). The contribution would be strengthened by more critical treatment of methodological confounds.
Reply: Added information in row 373-386 and row 755-758.
Comment: … collapsing multiple strategies (midpoint, endpoint, proportional reasoning, previous-number) into one “MEP” category reduces analytical granularity. The rationale should be explained more thoroughly, and alternative analyses considered.
Reply: This has been further explained in row 485-505.
Comment: Inter-rater reliability for coding strategies is not reported. If coding was done by multiple raters, please report reliability; if by a single rater, acknowledge this as a limitation.
Reply: Coding was done by a single rater per child/session. This has been further clarified in row 373-386 and row 763-764.
Comment: The control group did not engage in any alternative math activity. This makes it difficult to isolate the effects of number line training versus simple exposure to researcher attention. A stronger design would include an active control. This limitation should be addressed in the Discussion.
Reply: This has been further explained in row 758-762.
Comment: The study context (Swedish kindergartens, medium–high SES) should be more explicitly discussed in terms of cultural and demographic constraints. Results may not generalize to other contexts. Reply: Added information in row 785-790.
Comment: The qualitative findings (e.g., richer math talk in the physical condition) are insightful but somewhat separated from the main argument. Stronger integration with quantitative results would enrich the contribution.
Reply: Added information in row 614-618, 685-688, 703-713
Comment: The one-to-one researcher-led format limits transferability to typical classroom practice. Future directions could explore group-based adaptations.
Reply: This is discussed in row 770-784